# ACCELERATED RIEMANNIAN OPTIMIZATION: HANDLING CONSTRAINTS TO BOUND GEOMETRIC PENALTIES

## ABSTRACT

We propose a globally-accelerated, first-order method for the optimization of smooth and (strongly or not) geodesically-convex functions in Hadamard manifolds. Our algorithm enjoys the same convergence rates as Nesterov's accelerated gradient descent, up to a multiplicative geometric penalty and log factors. Crucially, we can enforce our method to stay within a compact set we define. Prior fully accelerated works *resort to assuming* that the iterates of their algorithms stay in some pre-specified compact set, except for two previous methods, whose applicability is limited to local optimization and to spaces of constant curvature, respectively. Achieving global and general Riemannian acceleration without iterates assumptively staying in the feasible set was asked as an open question in (Kim & Yang, 2022), which we solve for Hadamard manifolds. In our solution, we show that we can use a linearly convergent algorithm for constrained strongly g-convex smooth problems to implement a Riemannian inexact proximal point operator that we use as a subroutine, which is of independent interest.

## 1 INTRODUCTION

Riemannian optimization concerns the optimization of a function defined over a Riemannian manifold. It is motivated by constrained problems that can be naturally expressed on Riemannian manifolds allowing to exploit the geometric structure of the problem and effectively transforming it into an unconstrained one. Moreover, there are problems that are not convex in the Euclidean setting, but that when posed as problems over a manifold with the right metric, are convex when restricted to every geodesic, and this allows for fast optimization (Cruz Neto et al., 2006; Carvalho Bento & Melo, 2012; Bento et al., 2015; Allen-Zhu et al., 2018). That is, they are geodesically convex (g-convex) problems, cf. Definition 1.1. Some applications of Riemannian optimization in machine learning include dictionary learning (Cherian & Sra, 2017; Sun et al., 2017), robust covariance estimation in Gaussian distributions (Wiesel, 2012), Gaussian mixture models (Hosseini & Sra, 2015), operator scaling (Allen-Zhu et al., 2018), computation of Brascamp-Lieb constants (Bennett et al., 2008), Karcher mean (Zhang et al., 2016), Wasserstein Barycenters (Weber & Sra, 2017), low-rank matrix completion (Cambier & Absil, 2016; Heidel & Schulz, 2018; Mishra & Sepulchre, 2014; Tan et al., 2014; Vandereycken, 2013), optimization under orthogonality constraints (Edelman et al., 1998; Lezcano-Casado & Martínez-Rubio, 2019), and sparse principal component analysis (Genicot et al., 2015; Huang & Wei, 2019b; Jolliffe et al., 2003). The first seven problems are defined over Hadamard manifolds, which we consider in this work. In fact, the optimization in these cases is over symmetric spaces, which satisfy a property that one instance of our algorithm requires, cf. Theorem 2.4.

Riemannian optimization, whether under g-convexity or not, is an extensive and active area of research, for which one aspires to develop Riemannian optimization algorithms that share analogous properties to the more broadly studied Euclidean methods, such as the following kinds of Riemannian first-order methods: deterministic (Bento et al., 2017; Wei et al., 2016; Zhang & Sra, 2016), adaptive

---

Most of the notations in this work have a link to their definitions. For example, if you click or tap on any instance of $L$, you will jump to the place where it is defined as the smoothness constant of the function we consider in this work.

(Kasai et al., 2019), projection-free (Weber & Sra, 2017; 2019), saddle-point-escaping (Criscitiello & Boumal, 2019; Sun et al., 2019; Zhou et al., 2019; Criscitiello & Boumal, 2020), stochastic (Hosseini & Sra, 2017; Khuzani & Li, 2017; Tripuraneni et al., 2018), variance-reduced (Sato et al., 2017; 2019; Zhang et al., 2016), and min-max methods (Zhang et al., 2022), among others.

Riemannian generalizations to accelerated convex optimization are appealing due to their better convergence rates with respect to unaccelerated methods, specially in ill-conditioned problems. Acceleration in Euclidean convex optimization is a concept that has been broadly explored and has provided many different fast algorithms. A paradigmatic example is Nesterov's Accelerated Gradient Descent (AGD), cf. (Nesterov, 1983), which is considered the first general accelerated method, where the conjugate gradients method can be seen as an accelerated predecessor in a more limited scope (Martínez-Rubio, 2021). There have been recent efforts to better understand this phenomenon in the Euclidean case (Allen Zhu & Orecchia, 2017; Su et al., 2016; Drori & Teboulle, 2014; Wibisono et al., 2016; Diakonikolas & Orecchia, 2019; Joulani et al., 2020), which have yielded some fruitful techniques for the general development of methods and analyses. These techniques have allowed for a considerable number of new results going beyond the standard oracle model, convexity, or beyond first-order, in a wide variety of settings (Tseng, 2008; Beck & Teboulle, 2009; Wang et al., 2016a; Allen Zhu & Orecchia, 2015; Allen-Zhu, 2017; 2018; Carmon et al., 2017; Diakonikolas & Orecchia, 2018; Hinder et al., 2019; Gasnikov et al., 2019; Ivanova et al., 2021; Kamzolov & Gasnikov, 2020; Criado et al., 2021), among many others. There have been some efforts to achieve acceleration for Riemannian algorithms as generalizations of AGD, cf. Section 1.3. These works try to answer the following fundamental question:

*Can a Riemannian first-order method enjoy the same rates of convergence as Euclidean AGD?*

The question is posed under (possibly strongly) geodesic convexity and smoothness of the function to be optimized. And due to the lower bound in (Criscitiello & Boumal, 2021), we know the optimization must be under bounded geodesic curvature of the Riemannian manifold, and we might have to optimize over a bounded domain.

**Main result** In this work, we study the question above in the case of Hadamard manifolds $\mathcal{M}$ of bounded sectional curvature and provide an instance of our framework for a wide class of Hadamard manifolds. For a differentiable $f : \mathcal{M} \to \mathbb{R}$ with a global minimizer at $x^*$, let $x_0 \in \mathcal{M}$ be an initial point and $R$ be an upper bound on the distance $d(x_0, x^*)$. If $f$ is $L$-smooth and (possibly $\mu$-strongly) g-convex in a closed ball of center $x^*$ and radius $O(R)$, our algorithms obtain the same rates of convergence as AGD, up to logarithmic factors and up to a geometric penalty factor, cf. Theorem 2.4. See Table 1 for a succint comparison among accelerated algorithm and their rates. This algorithm is a consequence of the general framework we design:

*General accelerated scheme* Riemacon. Given a not necessarily accelerated, linearly-convergent subroutine for strongly g-convex smooth problems, constrained to a geodesically convex set $\mathcal{X}$, we design first-order algorithms that enjoy the same rates as AGD when approximating $\min_{x \in \mathcal{X}} f(x)$, up to logarithmic factors and up to a geometric penalty factor, where $f : \mathcal{N} \subset \mathcal{M} \to \mathbb{R}$ is a differentiable function that is smooth and g-convex (or strongly g-convex) in $\mathcal{X} \subset \mathcal{N}$, cf. Theorem 2.2.

Importantly, our algorithms obtain acceleration without an undesirable assumption that most previous works had to make: that the iterates of the algorithm stay inside of a pre-specified compact set without any mechanism for enforcing or guaranteeing this condition. To the best of our knowledge only two previous methods are able to deal with some form of constraints, and they apply to the limited settings of local optimization (Criscitiello & Boumal, 2021) and constant sectional curvature manifolds (Martínez-Rubio, 2021), respectively. Techniques in the rest of papers resort to assuming that the iterates of their algorithms are always feasible. Removing this condition in general, global, and fully accelerated methods was posed as an open question in (Kim & Yang, 2022), that we solve for the case of Hadamard manifolds. The difficulty of constraining problems in order to bound geometric penalties as well as the necessity of achieving this goal in order to provide full optimization guarantees with bounded geometric penalties is something that has also been noted in other kinds of Riemannian algorithms, cf. (Hosseini & Sra, 2020).

We develop new techniques on inexact proximal methods in Riemannian manifolds and show that with access to a (not necessarily accelerated) constrained linear subroutine for strongly g-convex and smooth problems, we can inexactly solve a proximal subproblem to enough accuracy so it can be used in our accelerated outer loop, in the spirit of other Euclidean algorithms like Catalyst (Lin

et al., 2017). After building this machinery, we show that we are able to implement an inexact ball optimization oracle, cf. (Carmon et al., 2020), as an instance of our solution. Crucially, the diameter $D$ of this ball depends on $R$ and the geometry only, so in particular it is independent on the condition number of $f$. We can use the linearly convergent algorithm in (Criscitiello & Boumal, 2021) for the implementation of the prox subroutine and we show that iterating the application of the ball optimization oracle leads to global accelerated convergence.

The question concerning whether there are Riemannian analogs to Nesterov's algorithm that enjoy similar rates is a question that, to the best of our knowledge, was first formulated in (Zhang & Sra, 2016). In particular, since Nesterov's AGD uses a proximal operator of a function's linearization, they ask whether there is a Riemannian analog to this operation that could be used to obtain accelerated rates in the Riemannian case. We show that, instead, a proximal step with respect to the *whole* function can be approximated efficiently in Hadamard manifolds and it can be used along with an accelerated outer loop. To the best of our knowledge, previously known Riemannian proximal methods either obtain asymptotic analyses, assume exact proximal computation, or work with approximate proximal operators by using different inexactness conditions as ours, and none of them show how to implement the proximal operators or obtain accelerated proximal point methods, cf. Section 1.3.

## 1.1 PRELIMINARIES

We provide definitions of Riemannian geometry concepts that we use in this work. The interested reader can refer to (Petersen, 2006; Bacák, 2014) for an in-depth review of this topic, but for this work the following notions will be enough. A Riemannian manifold $(\mathcal{M}, \mathfrak{g})$ is a real $C^\infty$ manifold $\mathcal{M}$ equipped with a metric $\mathfrak{g}$, which is a smoothly varying, i.e., $C^\infty$, inner product. For $x \in \mathcal{M}$, denote by $T_x\mathcal{M}$ the tangent space of $\mathcal{M}$ at $x$. For vectors $v, w \in T_x\mathcal{M}$, we denote the inner product of the metric by $\langle v, w \rangle_x$ and the norm it induces by $\|v\|_x \stackrel{\text{def}}{=} \sqrt{\langle v, v \rangle_x}$. Most of the time, the point $x$ is known from context, in which case we write $\langle v, w \rangle$ or $\|v\|$.

A geodesic of length $\ell$ is a curve $\gamma : [0, \ell] \to \mathcal{M}$ of unit speed that is locally distance minimizing. A uniquely geodesic space is a space such that for every two points there is one and only one geodesic that joins them. In such a case the exponential map $\text{Exp}_x : T_x\mathcal{M} \to \mathcal{M}$ and the inverse exponential map $\text{Log}_x : \mathcal{M} \to T_x\mathcal{M}$ are well defined for every pair of points, and are as follows. Given $x, y \in \mathcal{M}$, $v \in T_x\mathcal{M}$, and a geodesic $\gamma$ of length $\|v\|$ such that $\gamma(0) = x$, $\gamma(\|v\|) = y$, $\gamma'(0) = v/\|v\|$, we have that $\text{Exp}_x(v) = y$ and $\text{Log}_x(y) = v$. We denote by $d(x, y)$ the distance between $x$ and $y$, and note that it takes the same value as $\|\text{Log}_x(y)\|$. The manifold $\mathcal{M}$ comes with a natural parallel transport between vectors in different tangent spaces, that formally is defined from a way of identifying nearby tangent spaces, known as the Levi-Civita connection $\nabla$ (Levi-Civita, 1977). We use this parallel transport throughout this work.

Given a 2-dimensional subspace $V \subseteq T_x\mathcal{M}$ of the tangent space of a point $x$, the sectional curvature at $x$ with respect to $V$ is defined as the Gauss curvature, for the surface $\text{Exp}_x(V)$ at $x$. The Gauss curvature at a point $x$ can be defined as the product of the maximum and minimum curvatures of the curves resulting from intersecting the surface with planes that are normal to the surface at $x$. A Hadamard manifold is a complete simply connected Riemannian manifold whose sectional curvature is non-positive, like the hyperbolic space or the space of $n \times n$ symmetric positive definite matrices with the metric $\langle X, Y \rangle_A \stackrel{\text{def}}{=} \text{Tr}(A^{-1}XA^{-1}Y)$ where $X, Y$ are in the tangent space of $A$. Hadamard manifolds are uniquely geodesic. Note that in a general manifold $\text{Exp}_x(\cdot)$ might not be defined for each $v \in T_x\mathcal{M}$, but in a Hadamard manifold of dimension $n$, the exponential map at any point is a global diffeomorphism between $T_x\mathcal{M} \cong \mathbb{R}^n$ and the manifold, and so the exponential map is defined everywhere. We now proceed to define the main properties that will be assumed on our model for the function to be minimized and on the feasible set $\mathcal{X}$.

**Definition 1.1 (Geodesic Convexity and Smoothness).** Let $f : \mathcal{N} \subset \mathcal{M} \to \mathbb{R}$ be a differentiable function defined on an open set $\mathcal{N}$ contained in a Riemannian manifold $\mathcal{M}$. Given $L \geq \mu > 0$, we say that $f$ is *L-smooth* in a set $\mathcal{X} \subseteq \mathcal{N}$ if for any two points $x, y \in \mathcal{X}$, $f$ satisfies

$$f(y) \leq f(x) + \langle \nabla f(x), \text{Log}_x(y) \rangle + \frac{L}{2}d(x, y)^2.$$

Analogously, we say that $f$ is *μ-strongly g-convex* in $\mathcal{X}$, if for any two points $x, y \in \mathcal{X}$, we have

$$f(y) \geq f(x) + \langle \nabla f(x), \text{Log}_x(y) \rangle + \frac{\mu}{2}d(x, y)^2.$$

If the previous inequality is satisfied with $\mu = 0$, we say the function is *g-convex* in $\mathcal{X}$.

We present the following fact about the squared-distance function, when one of the arguments is fixed. The constants $\zeta_D, \delta_D$ below appear everywhere in Riemannian optimization because, among other things, Fact 1.2 yields Riemannian inequalities that are analogous to the equality in the Euclidean cosine law of a triangle, cf. Corollary C.3, and these inequalities have wide applicability in the analyses of Riemannian methods.

**Fact 1.2 (Local information of the squared-distance).** *Let $\mathcal{M}$ be a Riemannian manifold of sectional curvature bounded by $[\kappa_{\min}, \kappa_{\max}]$ that contains a uniquely g-convex set $\mathcal{X} \subset \mathcal{M}$ of diameter $D < \infty$. Then, given $x, y \in \mathcal{X}$ we have the following for the function $\Phi_x : \mathcal{M} \to \mathbb{R}$, $y \mapsto \frac{1}{2}d(x,y)^2$:*

$$\nabla \Phi_x(y) = -\operatorname{Log}_y(x) \qquad \text{and} \qquad \delta_D \|v\|^2 \leq \operatorname{Hess} \Phi_x(y)[v,v] \leq \zeta_D \|v\|^2,$$

*where*

$$\zeta_D \stackrel{\text{def}}{=} \begin{cases} D\sqrt{|\kappa_{\min}|} \coth(D\sqrt{|\kappa_{\min}|}) & \text{if } \kappa_{\min} \leq 0 \\ 1 & \text{if } \kappa_{\min} > 0 \end{cases},$$

*and*

$$\delta_D \stackrel{\text{def}}{=} \begin{cases} 1 & \text{if } \kappa_{\max} \leq 0 \\ D\sqrt{\kappa_{\max}} \cot(D\sqrt{\kappa_{\max}}) & \text{if } \kappa_{\max} > 0 \end{cases},$$

*In particular, $\Phi_x$ is $\delta_D$-strongly g-convex and $\zeta_D$-smooth in $\mathcal{X}$. See (Lezcano-Casado, 2020) for a proof.*

## 1.2 NOTATION.

Let $\mathcal{M}$ be a uniquely geodesic $n$-dimensional Riemannian manifold. Given points $x, y, z \in \mathcal{M}$, we abuse the notation and write $y$ in non-ambiguous and well-defined contexts in which we should write $\operatorname{Log}_x(y)$. For example, for $v \in T_x\mathcal{M}$ we have $\langle v, y - x \rangle = -\langle v, x - y \rangle = \langle v, \operatorname{Log}_x(y) - \operatorname{Log}_x(x) \rangle = \langle v, \operatorname{Log}_x(y) \rangle$; $\|v - y\| = \|v - \operatorname{Log}_x(y)\|$; $\|z - y\|_x = \|\operatorname{Log}_x(z) - \operatorname{Log}_x(y)\|$; and $\|y - x\|_x = \|\operatorname{Log}_x(y)\| = d(y,x)$. We denote by $\mathcal{X}$ a compact, uniquely geodesic g-convex set of diameter $D$ contained in an open set $\mathcal{N} \subset \mathcal{M}$ and we use $I_{\mathcal{X}}$ for the indicator function of $\mathcal{X}$, which is $0$ at points in $\mathcal{X}$ and $+\infty$ otherwise. For a vector $v \in T_y\mathcal{M}$, we use $\Gamma_y^x(v) \in T_x\mathcal{M}$ to denote the parallel transport of $v$ from $T_y\mathcal{M}$ to $T_x\mathcal{M}$ along the unique geodesic that connects $y$ to $x$. We call $f : \mathcal{N} \subset \mathcal{M} \to \mathbb{R}$ a differentiable $L$-smooth g-convex function we want to optimize. We use $\varepsilon$ to denote the approximation accuracy parameter, $x_0 \in \mathcal{X}$ for the initial point of our algorithms, and $\bar{R} \stackrel{\text{def}}{=} d(x_0, \bar{x}^*)$ for the initial distance to an arbitrary constrained minimizer $\bar{x}^* \in \arg\min_{x \in \mathcal{X}} f(x)$. We use $R$ for an upper bound on the initial distance $d(x_0, x^*)$ to an unconstrained minimizer $x^*$, if it exists. The big-$O$ notation $\widetilde{O}(\cdot)$ omits log factors. Note that in the setting of Hadamard manifolds, the bounds on the sectional curvature are $\kappa_{\min} \leq \kappa_{\max} \leq 0$. Hence for notational convenience, we define $\bar{\zeta} \stackrel{\text{def}}{=} \zeta_D = D\sqrt{|\kappa_{\min}|} \coth(D\sqrt{|\kappa_{\min}|}) \geq 1$, $\bar{\delta} \stackrel{\text{def}}{=} 1$, and similarly $\zeta \stackrel{\text{def}}{=} \zeta_R$ and $\delta \stackrel{\text{def}}{=} \delta_R = 1$. If $v \in T_x\mathcal{M}$, we use $\Pi_{\bar{B}(0,r)}(v) \in T_x\mathcal{M}$ for the projection of $v$ onto the closed ball with center at $0$ and radius $r$.

## 1.3 OUR RESULTS AND COMPARISONS WITH RELATED WORK

In this work, we optimize functions defined over Hadamard manifolds $\mathcal{M}$ of finite dimension $n$ and of sectional curvature bounded in $[\kappa_{\min}, \kappa_{\max}]$. As all previous related works discussed in the sequel, we assume that we can compute the exponential and inverse exponential maps, and parallel transport of vectors for our manifold. The differentiable function $f$ to be optimized is defined over an open set $\mathcal{N} \subset \mathcal{M}$ that contains a compact g-convex set $\mathcal{X}$ of finite diameter $D$. Our function $f$ is $L$-smooth and g-convex (or $\mu$-strongly g-convex) in $\mathcal{X}$ and we have access to it via a gradient oracle that can be queried at points in $\mathcal{X}$. For this setting, we show in Theorem 2.2 that with access to a (possibly unaccelerated) linearly convergent subroutine for g-strongly smooth problems in $\mathcal{X}$, the algorithms we propose find a point $y_T \in \mathcal{X}$ such that $f(y_T) - \min_{x \in \mathcal{X}} f(x) \leq \varepsilon$ after calling the gradient oracle and the subroutine the following number of times: $\widetilde{O}(\bar{\zeta}\sqrt{L\bar{R}^2/\varepsilon})$ for the g-convex case and $\widetilde{O}(\bar{\zeta}\sqrt{L/\mu} \log(\mu\bar{R}^2/\varepsilon))$ for the $\mu$-strongly g-convex case, where $\bar{R} \stackrel{\text{def}}{=} d(x_0, \bar{x}^*)$ and $x_0 \in \mathcal{X}$ is an initial point. Then in Theorem 2.4, we instantiate our algorithm with the method in (Criscitiello & Boumal, 2021) as subroutine and boost the convergence by

implementing and sequentially applying an inexact ball optimization oracle and we obtain the rates $\widetilde{O}(\zeta^2\sqrt{\zeta + LR^2/\varepsilon})$ and $\widetilde{O}(\zeta^2\sqrt{L/\mu}\log(\mu R^2/\varepsilon))$ where $R$ is a bound on the initial distance $d(x_0, x^*)$ to an unconstrained minimizer $x^*$. In sum, the algorithms enjoy the same rates as AGD in the Euclidean space up to a factor of $\zeta^2 = R^2\kappa_{\min}^2\coth^2(R\sqrt{|\kappa_{\min}|}) \leq (1 + R\cdot|\kappa_{\min}|)^2$ (our geometric penalty) and up to universal constants and log factors. Note that as the minimum curvature $\kappa_{\min}$ approaches 0 we have $\zeta \to 1$.

We emphasize that our Algorithm 1 only needs to query the gradient of $f$ at points in $\mathcal{X}$ and the $L$-smoothness and $\mu$-strong g-convexity of $f$ only need to hold in $\mathcal{X}$. This is relevant because in Riemannian manifolds the condition number $L/\mu$ in a set can increase with the size of the set, cf. (Martínez-Rubio, 2020, Proposition 27). Intuitively, although there are twice differentiable functions defined over the Euclidean space whose Hessian is constant everywhere, in other Riemannian cases the metric may preclude having such global condition and the larger the set is the larger the minimum possible condition number becomes. Compare this, for instance, with the bounds on the Hessian's eigenvalues of the squared-distance function in Fact 1.2, which are tight for spaces of constant curvature (Lezcano-Casado, 2020).

Now we proceed to compare our results with previous works. We have summarized most of the following discussion in Table 1. We include Nesterov's AGD in the table for comparison purposes[1]. There are some works on Riemannian acceleration that focus on empirical evaluation or that work under strong assumptions (Liu et al., 2017; Alimisis et al., 2019; Huang & Wei, 2019a; Alimisis et al., 2020; Lin et al., 2020), see (Martínez-Rubio, 2020) for instance for a discussion on these works. We focus the discussion on the most related work with guarantees. (Zhang & Sra, 2018) obtain an algorithm that, up to constants, achieves the same rates as AGD in the Euclidean space, for $L$-smooth and $\mu$-strongly g-convex functions but only *locally*, namely when the initial point starts in a small neighborhood $N$ of the minimizer $x^*$: a ball of radius $O((\mu/L)^{3/4})$ around it. (Ahn & Sra, 2020) generalize the previous algorithm and, by using similar ideas as in (Zhang & Sra, 2018) for estimating a lower bound on $f$, they adapt the algorithm to work globally, proving that it eventually decreases the objective as fast as AGD. However, as (Martínez-Rubio, 2020) noted, it takes as many iterations as the ones needed by Riemannian gradient descent (RGD) to reach the neighborhood of the previous algorithm. The latter work also noted that in fact RGD and the algorithm in (Zhang & Sra, 2018) can be run in parallel and combined to obtain the same convergence rates as in (Ahn & Sra, 2020), which suggested that for this technique, full acceleration with the rates of AGD only happens over the small neighborhood $N$ in (Zhang & Sra, 2018). Note however that (Ahn & Sra, 2020) show that their algorithm will decrease the function value faster than RGD, but this is not quantified. (Jin & Sra, 2021) developed a different framework, arising from (Ahn & Sra, 2020) but with the same guarantees for accelerated first-order methods. We do not feature it in the table. (Criscitiello & Boumal, 2021) showed, under mild assumptions, that in a ball of center $x \in \mathcal{M}$ and radius $O((\mu/L)^{1/2})$ containing $x^*$, the pullback function $f \circ \mathrm{Exp}_x : T_x\mathcal{M} \to \mathbb{R}$ is Euclidean, strongly convex, and smooth with condition number $O(L/\mu)$, so AGD yields local acceleration as well. In short, acceleration is possible in a small neighborhood because there the manifold is almost Euclidean and the geometric deformations are small in comparison to the curvature of the objective. These techniques fail for the g-convex case since the neighborhood becomes a point ($\mu/L = 0$).

Finding fully accelerated algorithms that are *global* presents a harder challenge. By a fully accelerated algorithm we mean one with rates with same dependence as AGD on $L$, $\varepsilon$, and if it applies, on $\mu$. (Martínez-Rubio, 2020) provided such algorithms for g-convex functions, strongly or not, defined over manifolds of constant sectional curvature and constrained to a ball of radius $R$. The convergence rates initially had large constants with respect to $R$ but were later improved, cf. Table 1. Kim & Yang (2022) designed global algorithms with the same rates as AGD up to universal constants and a factor of $\bar{\zeta}$, their geometric penalty. However, they need to assume that the iterates of their algorithm remain in their feasible set $\mathcal{X}$ and they point out on the necessity of removing such an assumption, which they leave as an open question. Our work solves this question for the case of Hadamard manifolds. In their technique, they show that they can use the structure of the accelerated scheme to *move* lower bound estimations on $f(x^*)$ from one particular tangent space to another without incurring extra errors, when the right Lyapunov function is used. By *moving* lower bounds here we mean finding

---

[1]Note that the original method in (Nesterov, 1983) needed to query the gradient of the function outside of the feasible set, and this was later improved to only require queries at feasible points (Nesterov, 2005) as in our work, hence our choice of citation in the table.

Table 1: Convergence rates of related works with provable guarantees for smooth problems over uniquely geodesic manifolds. Column **K?** refers to the supported values of the sectional curvature, **G?** to whether the algorithm is global (any initial distance to a minimizer is allowed). Here L and L′ mean they are local algorithms that require initial distance $O((L/\mu)^{-3/4})$ and $O((L/\mu)^{-1/2})$, respectively. Column **F?** refers to whether there is full acceleration, meaning dependence on $L$, $\mu$, and $\varepsilon$ like AGD up to possibly log factors. Column **C?** refers to whether the method can enforce some constraints. All methods require their iterates to be in some specified compact set, but the works with ✗ just assume the iterates will remain within the constraints. We use $\mathcal{W} \stackrel{\text{def}}{=} \sqrt{\frac{L}{\mu}} \log(\frac{LR^2}{\varepsilon})$. *A mild condition on the covariant derivative of the metric tensor is required.

| Method | g-convex | $\mu$-st. g-cvx | K? | G? | F? | C? |
|---|---|---|---|---|---|---|
| (Nesterov, 2005, AGD) | $O(\sqrt{\frac{LR^2}{\varepsilon}})$ | $O(\mathcal{W})$ | 0 | ✓ | ✓ | ✓ |
| (Zhang & Sra, 2018) | - | $O(\mathcal{W})$ | bounded | L | ✓ | ✗ |
| (Ahn & Sra, 2020) | - | $\widetilde{O}(\frac{L}{\mu} + \mathcal{W})$ | bounded | ✓ | ✗ | ✗ |
| (Martínez-Rubio, 2020) | $\widetilde{O}(\zeta\sqrt{\zeta + \frac{LR^2}{\varepsilon}})$ | $\widetilde{O}(\zeta \cdot \mathcal{W})$ | ctant.$\neq 0$ | ✓ | ✓ | ✓ |
| (Criscitiello & Boumal, 2021) | - | $O(\mathcal{W})$ | bounded* | L′ | ✓ | ✓ |
| (Kim & Yang, 2022) | $O(\zeta\sqrt{\frac{LR^2}{\varepsilon}})$ | $O(\zeta \cdot \mathcal{W})$ | bounded | ✓ | ✓ | ✗ |
| **Theorem 2.4** | $\widetilde{O}(\zeta^2\sqrt{\zeta + \frac{LR^2}{\varepsilon}})$ | $\widetilde{O}(\zeta^2 \cdot \mathcal{W})$ | Hadamard* | ✓ | ✓ | ✓ |

suitable lower bounds that are simple (a quadratic in their case), if pulled-back to one tangent space, if we start with a similar bound that is simple when pulled-back to another tangent space.

**Lower bounds.** In this paragraph, we omit constants depending on the curvature bounds in the big-$O$ notations for simplicity. (Hamilton & Moitra, 2021) proved an optimization lower bound showing that acceleration in Riemannian manifolds is harder than in the Euclidean space. (Criscitiello & Boumal, 2021) largely generalized their results. They essentially show that for a large family of Hadamard manifolds, there is a function that is smooth and strongly g-convex in a ball of radius $R$ that contains the minimizer $x^*$, and for which finding a point that is $R/5$ close to $x^*$ requires $\widetilde{\Omega}(R)$ calls to the gradient oracle. Note that these results do not preclude the existence of a fully accelerated algorithm with rates $\widetilde{O}(R)$+AGD rates, for instance. A similar hardness statement is provided for smooth and only g-convex functions. Also, reductions as in (Martínez-Rubio, 2020) evince this hardness is also present in this case.

**Handling constraints to bound geometric penalties.** In our algorithm and in all other known fully accelerated algorithms, learning rates depend on the diameter of the feasible set. This is natural: estimation errors due to geometric deformations depend on the diameter via the constants $\zeta_D, \delta_D$, the cosine-law inequalities Corollary C.3, or other analogous inequalities, and the algorithms take these errors into account. All other previous works are not able to deal with any constraints and hence they simply assume that the iterates of their algorithms stay within one such specified set, except for (Martínez-Rubio, 2020) and (Criscitiello & Boumal, 2021) that enforce a ball constraint, as we explained above. However, these two works have their applicability limited to spaces of constant curvature and to local optimization, respectively. Note that even if one could show that given a choice of learning rate, convergence implies that the iterates will remain in some compact set, then because the learning rates depend on the diameter of the set, and the diameter of the set would depend on the learning rates, one cannot conclude from this argument that the assumption these works make is going to be satisfied. In contrast, in this work, we design a general accelerated framework and an instance of it that keep the iterates bounded, effectively bounding geometric penalties while we do not need to resort to any other extra assumptions, solving the open question in (Kim & Yang, 2022).

**Riemannian proximal methods** There have been some works that study proximal methods in Riemannian manifolds, but most of them focus on asymptotic results or assume the proximal operator can be computed exactly (Wang et al., 2015; Bento et al., 2017; 2016; Khammahawong et al., 2021; Chang et al., 2021). The rest of these works study proximal point methods under different inexact versions of the proximal operator as ours and they do not show how to implement their inexact version

in applications, like in our case of smooth and g-convex optimization. In contrast, we implement the inexact proximal operator with a first-order method (Ahmadi & Khatibzadeh, 2014) provide a convergence analysis of an inexact proximal point method but when applied to optimization they assume the computation of the proximal operator is exact. (Tang & Huang, 2014) uses a different inexact condition and proves linear convergence, under a growth condition on $f$. (Wang et al., 2016b) obtains linear convergence of an inexact proximal point method under a different growth assumption on $f$ and under an absolute error condition on the proximal function.

## 2 ALGORITHMIC FRAMEWORK AND PSEUDOCODE

In this section, we present our **Riema**nnian **ac**celerated algorithm for **con**strained g-convex optimization, or Riemacon[2]. This is a general framework that we later instantiate to provide a full algorithm. Recall our abuse of notation for points $p \in \mathcal{M}$ to mean $\mathrm{Log}_q(p)$ in contexts in which one should place a vector in $T_q\mathcal{M}$ and note that in our algorithm $x_k$ and $y_k$ are points in $\mathcal{M}$ whereas $z_k^{x_k} \in T_{x_k}\mathcal{M}, z_k^{y_k}, \bar{z}_k^{y_k} \in T_{y_k}\mathcal{M}$.

---

**Algorithm 1** Riemacon: **Riema**nnian **Ac**celeration - **Con**strained g-Convex Optimization

**Input:** Feasible set $\mathcal{X}$. Initial point $x_0 \in \mathcal{X} \subset \mathcal{N}$. Diff. function $f : \mathcal{N} \subset \mathcal{M} \to \mathbb{R}$ for a Hadamard manifold $\mathcal{M}$ that is $L$-smooth and g-convex in $\mathcal{X}$. Optionally: final iteration $T$ or accuracy $\varepsilon$. If $\varepsilon$ is provided, compute the corresponding $T$, cf. Theorem 2.2.

    **Parameters:**
- Geometric penalty $\xi \stackrel{\mathrm{def}}{=} 4\zeta_{2D} - 3 \leq 8\bar{\zeta} - 3 = O(\bar{\zeta})$.
- Implicit Gradient Descent learning rate $\lambda \stackrel{\mathrm{def}}{=} \zeta_{2D}/L$.
- Mirror Descent learning rates $\eta_k \stackrel{\mathrm{def}}{=} a_k/\xi$.
- Proportionality constant in the proximal subproblem accuracies: $\Delta_k \stackrel{\mathrm{def}}{=} \frac{1}{(k+1)^2}$.

    **Definition:** (computation of this value is not needed)
- Prox. accuracies: $\sigma_k \stackrel{\mathrm{def}}{=} \frac{\Delta_k d(x_k, y_k^*)^2}{78\lambda}$ where $y_k^* \stackrel{\mathrm{def}}{=} \arg\min_{y \in \mathcal{X}}\{f(y) + \frac{1}{2\lambda}d(x_k, y)^2\}$.

---

1: $y_0 \leftarrow x_0; \quad A_0 \leftarrow 200\lambda\xi$
2: $z_0^{x_0} \leftarrow 0 \in T_{x_0}\mathcal{M}; \quad \bar{z}_0^{y_0} \leftarrow z_0^{y_0} \leftarrow 0 \in T_{y_0}\mathcal{M}$
3: **for** $k = 1$ **to** $T$ **do**
4:     $a_k \leftarrow 2\lambda\frac{k+32\xi}{5}$
5:     $A_k \leftarrow a_k/\xi + A_{k-1} = \sum_{i=1}^k a_i/\xi + A_0 = \lambda\left(\frac{k(k+1+64\xi)}{5\xi} + 200\xi\right)$
6:     $x_k \leftarrow \mathrm{Exp}_{y_{k-1}}(\frac{a_k}{A_{k-1}+a_k}\bar{z}_{k-1}^{y_{k-1}} + \frac{A_{k-1}}{A_{k-1}+a_k}y_{k-1}) = \mathrm{Exp}_{y_{k-1}}(\frac{a_k}{A_{k-1}+a_k}\bar{z}_{k-1}^{y_{k-1}})$     $\diamond$ Coupling
7:     $z_{k-1}^{x_k} \leftarrow \Gamma_{y_{k-1}}^{x_k}(\bar{z}_{k-1}^{y_{k-1}}) + \mathrm{Log}_{x_k}(y_{k-1}) = \mathrm{Log}_{x_k}(\mathrm{Exp}_{y_k}(\bar{z}_{k-1}^{y_{k-1}}))$
8:     $y_k \leftarrow \sigma_k$-minimizer of the proximal problem $\min_{y \in \mathcal{X}}\{f(y) + \frac{1}{2\lambda}d(x_k, y)^2\}$
9:     $v_k^x \leftarrow -\mathrm{Log}_{x_k}(y_k)/\lambda$                               $\diamond$ Approximate subgradient
10:     $z_k^{x_k} \leftarrow z_{k-1}^{x_k} - \eta_k v_k^x$                                  $\diamond$ Mirror Descent step
11:     $z_k^{y_k} \leftarrow \Gamma_{x_k}^{y_k}(z_k^{x_k}) + \mathrm{Log}_{y_k}(x_k)$                  $\diamond$ Moving the dual point to $T_{y_k}\mathcal{M}$
12:     $\bar{z}_k^{y_k} \leftarrow \Pi_{\bar{B}(0,D)}(z_k^{y_k}) \in T_{y_k}\mathcal{M}$     $\diamond$ Easy projection done so the dual point is not very far
13: **end for**
14: **return** $y_T$.

---

We start with an interpretation of our algorithm that helps understanding its high-level ideas. The following intends to be a qualitative explanation, and we refer to the pseudocode and the supplementary material for the exact descriptions and analysis. Euclidean accelerated algorithms can be interpreted, cf. (Allen Zhu & Orecchia, 2017), as a combination of a gradient descent (GD) algorithm and an online learning algorithm with losses being the affine lower bounds $f(x_k) + \langle \nabla f(x_k), \cdot - x_k \rangle$ we obtain on $f(\cdot)$ by applying convexity at some points $x_k$. That is, the latter builds a lower bound estimation on $f$. By selecting the next query to the gradient oracle as a cleverly picked convex combination of the predictions given by these two algorithms, one can show that the instantaneous

---

[2]Riemacon rhymes with "rima con" in Spanish.

regret of the online learning algorithm can be compensated by the local progress GD makes, which leads to accelerated convergence. In Riemannian optimization, there are two main obstacles. Firstly, the first-order approximations of $f$ at points $x_k$ yield functions that are affine but only with respect to their respective $T_{x_k}\mathcal{M}$, and so combining these lower bounds that are only simple in their tangent spaces makes obtaining good global estimations not simple. Secondly, when one obtains such global estimations, then one naturally incurs an instantaneous regret that is worse by a factor than is usual in Euclidean acceleration. This factor is a geometric constant depending on the diameter $D$ of a set $\mathcal{X}$ where the iterates and a (possibly constrained) minimizer lie. As a consequence, the learning rate of GD would need to be multiplicatively increased by such a constant with respect to the one of the online learning algorithm in order for the regret to still be compensated with the local progress of GD (and the rates worsen by this constant). But if we fix some $\mathcal{X}$ of finite diameter, because GD's learning rate is now larger, it is not clear how to keep the iterates in $\mathcal{X}$. And if we do not have the iterates in one such set $\mathcal{X}$, then our geometric penalties could grow arbitrarily.

We find the answer in implicit methods. An implicit Euclidean (sub)gradient descent step is one that computes, from a point $x_k \in \mathcal{X}$, another point $y_k^* = x_k - \lambda v_k \in \mathcal{X}$, where $v_k \in \partial(f + I_{\mathcal{X}})(y_k^*)$, is a subgradient of $f + I_{\mathcal{X}}$ at $y_k^*$. Intuitively, if we could implement a Riemannian version of an implicit GD step then it should be possible to still compensate the regret of the other algorithm and keep all the iterates in the set $\mathcal{X}$. Computing such an implicit step is computationally hard in general, but we show that approximating the proximal objective $h_k(y) \stackrel{\text{def}}{=} f(y) + \frac{1}{2\lambda}d(x_k, y)^2$ with enough accuracy yields an approximate subgradient that can be used to obtain an accelerated algorithm as well. In particular, we provide an accelerated scheme for which we show that the error incurred by the approximation of the subgradient can be bounded by some terms we can control, cf. Lemma A.2, namely a small term that appears in our Lyapunov function and also a term proportional to the squared norm of the approximated subgradient, which only increases the final convergence rates by a constant. This proximal approach works by exploiting the fact that the Riemannian Moreau envelop is convex in Hadamard manifolds (Azagra & Ferrera, 2005) and that the subproblem $h_k$, defined with our $\lambda = \zeta_{2D}/L$, is strongly g-convex and smooth with a condition number that only depends on the geometry. For this reason, a local algorithm like the one in (Criscitiello & Boumal, 2021) can be implemented in balls whose radius is independent on the condition number of $f$. Besides these steps, we use a coupling of the approximate implicit RGD and of a mirror descent (MD) algorithm, along with a technique in (Kim & Yang, 2022) to move dual points to the right tangent spaces without incurring extra geometric penalties, that we adapt to work with dual projections, cf. Lemma A.3. Importantly, the MD algorithm keeps the dual point close to the set $\mathcal{X}$ by using the projection in Line 12, which implies that the point $x_k$ is close to $\mathcal{X}$ as well, and this is crucial to keep low geometric penalties. This MD approach is a mix between follow-the-regularized-leader algorithms, that do not project the dual variable, and pure mirror descent algorithms that always project the dual variable. In the analysis, we note that partial projection also works, meaning that defining a new dual point that is closer to all of the points in the feasible set but without being a full projection leads to the same guarantees. Because we use the mirror descent lemma over $T_{y_k}\mathcal{M}$, what we described translates to: we can project the dual $z_k^{y_k}$ onto a ball defined on $T_{y_k}\mathcal{M}$ that contains the pulled-back set $\text{Log}_{y_k}(\mathcal{X})$ and by means of that trick we can keep the iterates $x_k$ close to $\mathcal{X}$. And at the same time, the point for which we prove guarantees, namely $y_k$, is always in $\mathcal{X}$.

Finally, we instantiate our subroutine with the algorithm in (Criscitiello & Boumal, 2021), in balls of radius independent on the condition number of $f$ and show in Theorem 2.4 that if we iterate this approximate implementation of a ball optimization oracle, we obtain convergence at a globally accelerated rate. We note (Zhang & Sra, 2016) also provided a claimed linearly convergent algorithm for constrained strongly g-convex smooth problems, and thus in principle it could be used for our subroutine. Unfortunately, we noticed that the proof is flawed when the optimization is constrained.

We leave the proofs of most of our results to the supplementary material and state our main theorems below. Using the insights explained above, we show the following inequality on $\psi_k$, defined below, that will be used as a Lyapunov function to prove the convergence rates of Algorithm 1.

**Proposition 2.1.** [↓] *By using the notation of Algorithm 1, let*

$$\psi_k \stackrel{\text{def}}{=} A_k(f(y_k) - f(\bar{x}^*)) + \frac{1}{2}\|z_k^{y_k} - \text{Log}_{y_k}(\bar{x}^*)\|_{y_k}^2 + \frac{\xi - 1}{2}\|z_k^{y_k}\|_{y_k}^2.$$

*Then, for all $k \geq 1$, we have $(1 - \Delta_k)\psi_k \leq \psi_{k-1}$.*

Finally, we can state our theorem for the optimization of $L$-smooth and g-convex functions.

**Theorem 2.2.** [↓] *Let $\mathcal{M}$ be a finite-dimensional Hadamard manifold of bounded sectional curvature, and consider $f : \mathcal{N} \subset \mathcal{M} \to \mathbb{R}$ be an L-smooth and g-convex differentiable function in a compact g-convex set $\mathcal{X} \subset \mathcal{N}$ of diameter $D$, $\bar{x}^* \in \arg\min_{x \in \mathcal{X}} f(x)$, and $\bar{R} \stackrel{\text{def}}{=} d(x_0, \bar{x}^*)$. For any $\varepsilon > 0$, Algorithm 1 yields an $\varepsilon$-minimizer $y_T \in \mathcal{X}$ after $T = O(\bar{\zeta}\sqrt{\frac{L\bar{R}^2}{\varepsilon}})$ iterations. If the function is $\mu$-strongly convex then, via a sequence of restarts, we converge in $O(\bar{\zeta}\sqrt{\frac{L}{\mu}}\log(\frac{\mu\bar{R}^2}{\varepsilon}))$ iterations.*

We note that a straightforward corollary from our results is that if we can compute the exact Riemannian proximal point operator and we use it as the implicit gradient descent step in Line 8 of Algorithm 1, then the method is an accelerated proximal point method. One such Riemannian algorithm was previously unknown in the literature as well. Finally, we instantiate Algorithm 1 to implement approximate ball optimization oracles in an accelerated way. We show that applying these oracles sequentially leads to global accelerated convergence. Moreover, we show that the iterates do not get farther than $2R$ from $x^*$, which ultimately leads to the geometric penalty being a function of $\zeta$ and not on the condition number of $f$. For the subroutine in Line 8 of Algorithm 1, we use the algorithm in (Criscitiello & Boumal, 2021, Section 6), and for that we require the following.

**Assumption 2.3.** Let $\mathfrak{R}$ be the curvature tensor of a Riemannian manifold $\mathcal{M}$. Its covariant derivative is $\nabla\mathfrak{R} = 0$.

We note that locally symmetric manifolds, like $\mathrm{SO}(n)$, the SPD matrix manifold, the Grasmannian manifold, manifolds of constant sectional curvature are all manifolds such that $\nabla\mathfrak{R} = 0$. We argue that this assumption is mild, since in particular these manifolds cover all of the applications in Section 1.

**Theorem 2.4.** [↓] *Let $\mathcal{M}$ be a finite-dimensional Hadamard manifold of bounded sectional curvature satisfying Assumption 2.3. Consider $f : \mathcal{N} \subset \mathcal{M} \to \mathbb{R}$ be an L-smooth and $\mu$-strongly g-convex differentiable function in $\bar{B}(x^*, 3R)$, where $x^*$ is its global minimizer and where $R \geq d(x_0, x^*)$ for an initial point $x_0$. For any $\varepsilon > 0$, Algorithm 2 yields an $\varepsilon$-minimizer after $\widetilde{O}(\zeta^2\sqrt{L/\mu}\log(LR^2/\varepsilon))$ calls to the gradient oracle of $f$. By using regularization, this algorithm $\varepsilon$-minimizes the g-convex case ($\mu = 0$) after $\widetilde{O}(\zeta^2\sqrt{\zeta + LR^2/\varepsilon})$ gradient oracle calls.*

---

**Algorithm 2** Ball Optimization Boosting of a Riemacon instance (Algorithm 1)

---

**Input:** Differentiable function $f : \mathcal{N} \subset \mathcal{M} \to \mathbb{R}$ that is L-smooth and $\mu$-strongly g-convex in $\bar{B}(x^*, 3R) \subset \mathcal{N}$; initial point $x_0 \in \mathcal{N}$; bound $R \geq d(x_0, x^*)$; constant $F$ from Assumption 2.3; accuracy $\varepsilon$.

---

1: **if** $2R \leq (46R|\kappa_{\min}|\zeta_{2R})^{-1}$ **then return** RiemaconSC($\bar{B}(x_0, R), x_0, f, \varepsilon$)
2: Compute $D$ such that $D = (46R|\kappa_{\min}|\zeta_D)^{-1}$. Alternatively, make $D \leftarrow (70R|\kappa_{\min}|)^{-1}$.
3: $T \leftarrow \lceil\frac{4R}{D}\ln(\frac{LR^2}{\varepsilon})\rceil$; $\varepsilon' \leftarrow \min\{\frac{D\varepsilon}{8R}, \frac{\mu R^2}{2T^2}\}$
4: **for** $k = 1$ **to** $T$ **do**
5: $\quad \mathcal{X}_k \leftarrow \bar{B}(x_{k-1}, D/2)$
6: $\quad x_k \leftarrow$ RiemaconSC($\mathcal{X}_k, x_{k-1}, f, \varepsilon'$) $\qquad \diamond$ (Criscitiello & Boumal, 2021) as subroutine
7: $\quad \diamond$ RiemaconSC is the strongly convex version of Algorithm 1 in Theorem 2.2 (cf. its proof).
8: **end for**
9: **return** $x_T$.

---

## 3 CONCLUSION AND FUTURE DIRECTIONS

In this work, we pursued an approach that, by designing and making use of inexact Riemannian proximal methods, yielded accelerated optimization algorithms. Consequently we were able to work without an undesirable assumption that most previous methods required, whose potential satisfiability is not clear: that the iterates stay in certain specified geodesically-convex set without enforcing them to be in the set. A future direction of research is the study of whether there are algorithms like ours that incur even lower geometric penalties or that do not incur $\log(1/\varepsilon)$ factors. Another interesting direction consists of studying generalizations of our approach to more general manifolds, namely the full Hadamard case, and manifolds of non-negative or even of bounded sectional cuvature.

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
