# OpenReview forum: "Accelerated Riemannian Optimization: Handling Constraints to Bound Geometric Penalties"
_ICLR.cc/2023/Conference — Submitted to ICLR 2023_

### Official Review · Reviewer_S9aC · 2022-10-24

**Confidence:** 1
**Correctness:** 3
**Technical Novelty And Significance:** 3
**Empirical Novelty And Significance:** 2
**Recommendation:** 5

**Clarity, Quality, Novelty And Reproducibility:**

This paper is very hard to understand. It requires much knowledge of the background. Since the authors do not provide any concrete examples, the paper is really not clear to me. I believe the quality of this paper is good. Besides, given its strong theoretical results, the novelty is quite strong.

**Strength And Weaknesses:**

Strength:
1. The designed algorithm looks very interesting because it seems to be a combination of the extension of Nesterov's acceleration gradient method on the Riemannian manifold and the mirror descent algorithm.
2. The theoretical results are very strong since the most accelerated algorithms can only be proved to enjoy the same convergence rate as standard GD, even though they perform much better than GD in practice.

Weaknesses:
1. This paper is very hard to understand. The authors should give more background on the problem and show some examples. At least there is no formally mathematical description of the problem. Besides, For each step in the algorithm, the authors should give more intuition to help us to understand it.
2. No concrete example or application is mentioned in this paper. Even though it is a theoretical paper, the author should mention at least one concrete example that satisfies all the assumptions and can be solved well using the proposed algorithm.
3. No simulation results. The author should compare their method with, for example, a non-accelerated first-order method to show the strength of acceleration.  It will be great if a better convergence rate is observed.

**Summary Of The Paper:**

In this paper, the authors propose an accelerated first-order method for the optimization of the Hadamard manifold. They theoretically show that the proposed algorithm enjoys the same convergence rate as Nesterov's accelerated gradient descent method in Euclidian space, under the assumption that the objective function is geodesically-convex. Besides, they can apply the linearly convergent algorithm to solve the subproblem in each iteration, which is a constrained strongly geodesical-convex smooth problem.

**Summary Of The Review:**

This paper should be a very good theoretical paper, even though I can not understand all the details. However, owing to the lack of concrete applications and numerical experiments, I think it may not be suitable for this conference, so I recommend not accepting this paper. I further recommend the authors submit it to a journal directly, given its amazing theoretical contributions.

---

> ### Author Response · Authors · 2022-11-18
> **response to reviewer**
>
> > No concrete example or application is mentioned in this paper. Even though it is a theoretical paper, the author should mention at least one concrete example that satisfies all the assumptions and can be solved well using the proposed algorithm.
>
> We cited several Hadamard problems to which our results apply to. Assumption 5 is satisfied as we mentioned. But we should emphasize and make more clear that all the Hadamard convex problems we cited can be tackled with our algorithms. We thank the reviewer for pointing out that the writing was not clear and we will change the phrasing.
>
>
> > mathematical description is lacking
>
> We did include a mathematical description of the problem and the solution.
>
> "For a differentiable $f:\mathcal{M}\to\mathbb{R}$ with a global minimizer at $x^\ast$, let $x_0 \in \mathcal{M}$ be an initial point and $R$ be an upper bound on the distance $d(x_0, x^\ast)$. If $f$ is $L$-smooth and (possibly $\mu$-strongly) g-convex in a closed ball of center $x^ast$ and radius $O(R)$, our algorithms obtain the same rates of convergence as AGD, up to logarithmic factors and up to a geometric penalty factor, cf. Theorem 2.4"
>
>
> > This paper should be a very good theoretical paper, even though I can not understand all the details. However, owing to the lack of concrete applications and numerical experiments, I think it may not be suitable for this conference, so I recommend not accepting this paper. I further recommend the authors submit it to a journal directly, given its amazing theoretical contributions.
>
> We note that we mentioned several applications with its citatations and we cited several works that spell out the details of these applications. See for example the monograph that we cited from Hosseini & Sra 2020.
>
> > numerical experiments
>
> As we said to the first reviewer:
>
> This paper is a theoretical study that focuses on designing algorithms that are provably accelerated and work under realistic assumptions. Previous accelerated methods make some assumptions that cannot be shown to be satisfied and it cannot be compared with our algorithm. Comparison with unaccelerated methods is possible. While we believe that experiments to check that these methods work well with real problems is a worthwhile goal, the focus of these paper was foundational and a thorough empirical comparison requires a scientific effort that would require a different paper. Besides, we believe that our theorems with provable guarantees say much more than a few plots with arbitrarily chosen problems.
>
> There are several subcomunities in ICLR and it seems to us that people outside of the theory subcommunity do not appreciate the value of bringing new provable ideas to the table. Note that ICLR is a conference with a strong optimization community to which these theoretical works are relevant. As an example, here are a couple of accepted papers of these kind from last year (i.e., strong theory results with no experiments) https://openreview.net/forum?id=sA4qIu3zv6v https://openreview.net/forum?id=J4iSIR9fhY0 (accepted as spotlight) and these are a couple of submissions with good scores from this year https://openreview.net/forum?id=yYbhKqdi7Hz https://openreview.net/forum?id=cB4N3G5udUS, just to name a few.

---

### Official Review · Reviewer_qdaa · 2022-10-24

**Confidence:** 3
**Clarity, Quality, Novelty And Reproducibility:** Very clear, only minor flaws.
**Correctness:** 4
**Technical Novelty And Significance:** 3
**Empirical Novelty And Significance:** 3
**Recommendation:** 6

**Strength And Weaknesses:**

Strength: This paper proposed a globally-accelerated, first-order method for the optimization of smooth and (strongly or not) geodesically-convex functions in Hadamard manifolds. Technically strong, highly general results, advanced techniques

**Summary Of The Paper:**

This paper proposed a globally-accelerated, first-order method for the optimization of smooth and (strongly or not) geodesically-convex functions in Hadamard manifolds.It enjoys the same convergence rates as Nesterov’s accelerated gradient descent.The most improvement is that this method without the assumption that the iterates of their algorithms stay in some pre-specifified compact set.The theoretical work is comprehensive.

**Summary Of The Review:**

Although the paper is theoretically sound, there are still some questions need to be discussed in this paper:
1.	In written. Although we know what problem we study, It would be better to add  mathematical description .
2.	A mistake.In notation,  ||v-y||=||v-\log_x(y)|| may be ||v-y||=||\log_x(y)-y|| .
3.	About workload. The authors use a linearly convergent algorithm for constrained strongly g-convex smooth problems to implement the Line 8 of Algorithm 1.But from my point of view, the Riemacon instance for g-convex function is also important.Perhaps the authors want to add this in future work.

---

> ### Author Response · Authors · 2022-11-18
> **response to reviewer**
>
> > 1. In written. Although we know what problem we study, It would be better to add mathematical description .
>
> We would appreciate if the reviewer could ellaborate on this indicating what mathematical description they felt it was lacking, so we can improve the writing of our work. The main result section summarizes the result with a mathematical description (which includes the problem). Similarly for the theorem statements.
>
> "For a differentiable $f:\mathcal{M}\to\mathbb{R}$ with a global minimizer at $x^\ast$, let $x_0 \in \mathcal{M}$ be an initial point and $R$ be an upper bound on the distance $d(x_0, x^ast)$. If $f$ is $L$-smooth and (possibly $\mu$-strongly) g-convex in a closed ball of center $x^ast$ and radius $O(R)$, our algorithms obtain the same rates of convergence as AGD, up to logarithmic factors and up to a geometric penalty factor, cf. Theorem 2.4"
>
> > 2. A mistake.In notation, ||v-y||=||v-\log_x(y)|| may be ||v-y||=||\log_x(y)-y||.
>
> That line in the notation section is correct. The notation specifies that if you place a point $y \in \mathcal{M}$ in some place where you should place a vector in $T_x\mathcal{M}$, then that notation means the vector in $T_x\mathcal{M}$ that takes you to the point $y \in \mathcal{M}$. In $|v-y|$, since $v \in T_x\mathcal{M}$, then $y$ must be a vector in $T_x\mathcal{M}$ and it is thus $\operatorname{Log}_x(y)$.
>
>
> > 3. About workload. The authors use a linearly convergent algorithm for constrained strongly g-convex smooth problems to implement the Line 8 of Algorithm 1.But from my point of view, the Riemacon instance for g-convex function is also important.Perhaps the authors want to add this in future work.
>
> We are not sure what the reviewer refers to. We already cover all cases:
>
> + Line 8 is always a strongly g-convex problem because $(1/2\lambda)d(x_k, y)^2$ is added, regardless of whether $f$ is strongly g-convex or not.
>
> + On the other hand, all our algorithms (the framework, the instance, and the boosted algorithm with a ball optimization oracle) have their versions for both g-convex and for strongly g-convex problems by means of regularization or restarts. For example, algorithm 1 Riemacon is designed for g-convex problems and we show in Theorem 2.2 that the same algorithm, applied with restarts, optimizes strongly g-convex problems at the accelerated rate. See the statement of both of our theorems, and Table 1

---

### Official Review · Reviewer_E5FH · 2022-11-03

**Confidence:** 4
**Correctness:** 3
**Technical Novelty And Significance:** 3
**Empirical Novelty And Significance:** 2
**Recommendation:** 3

**Clarity, Quality, Novelty And Reproducibility:**

A lot of discussion is at high level. As the paper claims to solve an open problem, it would have made sense to compare a bit more deeply on what were the challenges which others could not solve but was solved in the paper. E.g., which particular step made the crucial difference. Those details are missing.

**Strength And Weaknesses:**

Strengths:
It is an interesting piece of research which aims to solve an important theoretical problem in Riemannian optimization. The developments seem to be on solid foundation.

Weakness:
I find the paper far from complete. I would have liked to see some numerical experiments to see how useful Algorithm 1 is in practice to other methods. Without this, the paper looks very incomplete for the audience of this conference.


**Summary Of The Paper:**

The paper presents a theoretical result on accelerated methods for Riemannian optimization. In particular, it proposes  general Riemannian acceleration without iterates assumptively staying in the feasible set (which is an interesting open question posed earlier).


**Summary Of The Review:**

I find the work interesting but very far from complete for the audience of this conference.

---

> ### Author Response · Authors · 2022-11-18
> **response to reviewer**
>
> > I find the paper far from complete. I would have liked to see some numerical experiments to see how useful Algorithm 1 is in practice to other methods. Without this, the paper looks very incomplete for the audience of this conference.
>
> This paper is a theoretical study that focuses on designing algorithms that are provably accelerated and work under realistic assumptions. Previous accelerated methods make some assumptions that cannot be shown to be satisfied and it cannot be compared with our algorithm. Comparison with unaccelerated methods is possible. While we believe that experiments to check that these methods work well with real problems is a worthwhile goal, the focus of these paper was foundational and a thorough empirical comparison requires a scientific effort that would require a different paper. Besides, we believe that our theorems with provable guarantees say much more than a few plots with arbitrarily chosen problems.
>
> There are several subcomunities in ICLR and it seems to us that people outside of the theory subcommunity do not appreciate the value of bringing new provable ideas to the table. Note that ICLR is a conference with a strong optimization community to which these theoretical works are relevant. As an example, here are a couple of accepted papers of these kind from last year (i.e., strong theory results with no experiments) https://openreview.net/forum?id=sA4qIu3zv6v https://openreview.net/forum?id=J4iSIR9fhY0 (accepted as spotlight) and these are a couple of submissions with good scores from this year https://openreview.net/forum?id=yYbhKqdi7Hz https://openreview.net/forum?id=cB4N3G5udUS, just to name a few.
>
>
> > A lot of discussion is at high level. As the paper claims to solve an open problem, it would have made sense to compare a bit more deeply on what were the challenges which others could not solve but was solved in the paper. E.g., which particular step made the crucial difference. Those details are missing.
>
> Please note that all the low level discussions are in the paper. We explain our results and comparisons with related work in section 1, we discuss the techniques in section 2 and then we have all the technical proofs in the appendix, which are very detailed.
>
> > As the paper claims to solve an open problem, it would have made sense to compare a bit more deeply on what were the challenges which others could not solve but was solved in the paper. E.g., which particular step made the crucial difference. Those details are missing.
>
> We gave a deep comparison in the related work section about how we differ from previous works. And we talked about all the new things that make our solution work: proximal point methods in Riemannian optimization, ball optimization oracle, how we handle constraints to bound geometric penalties, among other things
>
> And we commented on key steps that made the crucial difference in the discussion after the pseudocode of algorithm 1, how methods try to combine gradient descent with a Riemannian version mirror descent but they get a greater regret so in order to compensate it with gradient descent they need the learning rate of the latter to be larger, not allowing them to guarantee they stay in the set. And we solve this by using implicit methods. Implicit gradient descent will stay in the set despite that the learning rate is larger.
>
> Also, the solution is not incremental, there is not "a particular single step that made the crucial difference".

---

### Decision · Program_Chairs · 2023-01-20

**Decision:**

Reject

**Justification For Why Not Higher Score:**

I quite like this paper myself but due to agreement from reviewers finding it incomplete without some empirical validation, do not choose to overturn their view

**Justification For Why Not Lower Score:**

n/a

**Metareview: Summary, Strengths And Weaknesses:**

This paper provides, from my view, strong theoretical contributions toward understanding accelerated optimization methods in the Hadamard manifold setup. The paper is quite technical, and reviewers found that the exposition can be improved and the contributions clarified. While I find many good ideas in this paper especially as it primarily targets theoretical contributions, the reviewers are in consensus that the paper is incomplete, especially with zero simulation results. I appreciate the space for theory papers in ICLR but also understand that even without deep empirical studies, submissions that at least demonstrate the implementability of an algorithm or its performance on some simulated benchmarks make the contributions substantially easier to appreciate.